# AgriPath: A Systematic Exploration of Architectural Trade-offs for Crop Disease Classification

**Hamza Mooraj**  *hhmooraj@gmail.com*
*School of Mathematical and Computer Sciences*
*Heriot-Watt University*

**George Pantazopoulos**  *gmp2000@hw.ac.uk*
*School of Mathematical and Computer Sciences*
*Heriot-Watt University*

**Alessandro Suglia**  *asuglia@ed.ac.uk*
*Institute for Language, Cognition and Computation, School of Informatics*
*University of Edinburgh*

**Reviewed on OpenReview:** *https://openreview.net/forum?id=5UI1wrq5pS*

## Abstract

Reliable crop disease detection requires models that perform consistently across diverse acquisition conditions, yet existing evaluations often focus on single architectural families or lab-generated datasets. This work presents a systematic empirical comparison of three model paradigms for fine-grained crop disease classification: Convolutional Neural Networks (CNNs), contrastive Vision–Language Models (VLMs), and generative VLMs. To enable controlled analysis of domain effects, we introduce *AgriPath-LF16*, a benchmark of 111k images spanning 16 crops and 41 diseases with explicit separation between laboratory and field imagery, alongside a balanced 30k subset for standardised training and evaluation. We train and evaluate all models under unified protocols across full, lab-only, and field-only training regimes using macro-F1 and Parse Success Rate (PSR) to account for generative reliability (i.e., output parsability measured via PSR). The results reveal distinct performance profiles: CNN-based models achieve the strongest in-domain performance but exhibit pronounced degradation under domain shift; contrastive VLMs provide a robust and parameter-efficient alternative with competitive cross-domain performance; generative VLMs demonstrate the strongest resilience to distributional variation, albeit with additional failure modes stemming from free-text generation. These findings highlight that architectural choice should be guided by deployment context rather than aggregate performance alone. The AgriPath-LF16 benchmark and accompanying codebase are publicly available at the following links: *Huggingface Dataset*; *GitHub Repository*.

## 1 Introduction

Accurate crop disease detection is critical for mitigating yield loss and supporting food security under increasingly variable environmental conditions. While automated systems can scale diagnostic support, their performance and reliability depend on the context of deployment. Understanding how different model architectures behave in a variety of agricultural contexts remains essential for deploying computer vision systems in this field.

Despite progress, most evaluations focus on narrow architectural families or curated lab imagery. Convolutional Neural Networks (CNNs) (He et al., 2015), contrastive Vision-Language Models (VLMs) (Radford et al., 2021; Zhai et al., 2023), and generative VLMs (Bai et al., 2025; Marafioti et al., 2025) possess distinct inductive biases and pre-training regimes, yet their comparative performance in complex field environments

remains underexplored. Furthermore, existing datasets are frequently crop-specific, domain-imbalanced, and often lack explicit separation between acquisition domains (lab and field). This lack of granularity obscures whether performance reflects genuine visual understanding or domain-specific overfitting, complicating the evaluation of deployment trade-offs.

This work provides a systematic empirical study of model architectures for fine-grained crop disease classification. The primary contributions are:

(1) *AgriPath-LF16:* A domain-aware benchmark featuring 111k images spanning 16 crops and 41 diseases with explicit separation between lab and field acquisition domains. A balanced 30k subset is constructed to support controlled training and evaluation. The dataset is publicly released to facilitate reproducible research.

(2) *Multi-Paradigm Comparison:* A unified comparison of CNNs, contrastive VLMs, and generative VLMs is conducted under a shared experimental protocol, incorporating a metric suite of macro-F1 across domains and Parse Success Rate (PSR), which measures the proportion of model outputs that can be mapped to valid crop–disease labels, capturing structural reliability of generative predictions alongside classification performance; enabling direct comparison between deterministic classifiers and free-text generative models.

(3) *Deployment Context Analysis:* Model performance is evaluated under domain-focused training regimes to characterise robustness, in-domain performance, and cross-domain transfer. The analysis highlights distinct performance profiles, clarifying which architectures best suit specific agricultural deployment contexts.

## 2 Related Work

Advances in computer vision have enabled widespread use of deep learning for image recognition, object detection, and segmentation across various domains. This section surveys prior datasets for crop disease classification, the implications of different computer vision architectures in this task, and outlines the remaining limitations that motivate this study.

### 2.1 Datasets for Crop Disease Classification

Public datasets for plant disease detection remain limited in scale, environmental diversity, and domain coverage. The widely used PlantVillage dataset introduced by Hughes & Salathe (2016) contains 54,306 images across 14 crops and 38 diseases but is almost entirely composed of lab-captured images with uniform backgrounds, restricting its suitability for evaluating models under real-world conditions. PlantDoc (Singh et al., 2020) provides field-based imagery but at a substantially smaller scale (2,922 samples), limiting its utility for training high-capacity models.

Other datasets such as Plant Pathology 2021 (Thapa et al., 2021) and CornNLB (Wiesner-Hanks & Brahimi, 2019) offer high-quality images but focus narrowly on specific crops and disease categories. Collectively, existing resources exhibit three limitations: (i) strong bias toward lab imagery, (ii) restricted crop and disease diversity, and (iii) insufficient paired lab–field coverage for assessing robustness under distributional shift. To address these gaps, this work introduces AgriPath-LF16, a larger-scale benchmark spanning 16 crops and 41 diseases with explicit separation between lab and field imagery, enabling controlled analysis of performance across heterogeneous deployment contexts.

### 2.2 CNN-Based Approaches

CNNs have long been the dominant approach for plant disease classification due to their strong inductive biases for local spatial pattern extraction (He et al., 2015). Prior studies demonstrate high performance when training and evaluation occur within the same domain: for example, Islam et al. (2023) report a 98.99% accuracy for ResNet-50 on a PlantVillage subset, and similar lab-based results are found across rice and cotton datasets in K.M. et al. (2023); Hassan et al. (2022). However, several works highlight severe degradation under distribution shift. Mohanty et al. (2016) show accuracy falling to just above 31% when PlantVillage-trained models were tested on field images.

These findings indicate that CNNs tend to learn representations tightly coupled to their training distribution, resulting in performance variation under heterogeneous acquisition conditions. This trade-off between peak in-domain accuracy and cross-context reliability motivates a controlled comparison against architectures with broader pre-training regimes. Accordingly, this work evaluates CNNs alongside contrastive and generative vision–language models under unified protocols to characterise their relative strengths across diverse deployment contexts.

### 2.3 Vision–Language Models for Agricultural Tasks

Vision–Language Models (VLMs) combine large-scale visual and textual pre-training, enabling models to align image representations with language-conditioned outputs (Radford et al., 2021). Their application to crop disease detection remains relatively recent. The VLCD framework (Zhou et al., 2024) integrates visual features with textual disease descriptions, reporting promising results on grape leaf diseases, although its single-crop focus limits broader generalisability.

More recent work, such as AgroGPT (Awais et al., 2024), extends multimodal LLMs toward expert-style reasoning and visual question answering (VQA) in agricultural contexts through the AgroEvals benchmark, emphasising conversational capability and advisory interaction. However, their primary focus does not lie in fine-grained classification across diverse crops and acquisition conditions. Emerging benchmarks such as MIRAGE, introduced by Dongre et al. (2025), evaluate VLMs on information-seeking behaviour and multimodal dialogue with agricultural experts. Similar to AgroGPT, such benchmarks presuppose that models possess reliable visual understanding.

To address these limitations, this work provides a unified empirical comparison of model architectures under various training and evaluation regimes. By examining performance across acquisition contexts, it assesses whether the visual capabilities assumed in advisory and conversational benchmarks translate into reliable fine-grained classification under heterogeneous deployment conditions.

## 3 Methodology

### 3.1 Dataset

This study introduces AgriPath-LF16, a large-scale dataset containing 111,307 images across 16 crops, 41 diseases, and 65 crop–disease pairs. AgriPath-LF16 was compiled from multiple public datasets (Appendix Table 3) where samples were mapped into a unified crop–disease ontology, with each class defined as a canonical crop–disease pair. The dataset spans two acquisition domains: controlled laboratory conditions and natural field environments; each image is annotated with its acquisition domain (lab or field) based on its source dataset. Lab images exhibit uniform backgrounds and consistent lighting, whereas field images contain cluttered scenes and variable conditions (Figure 1).

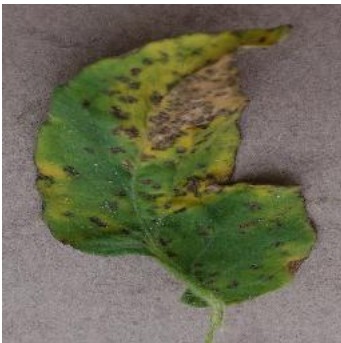 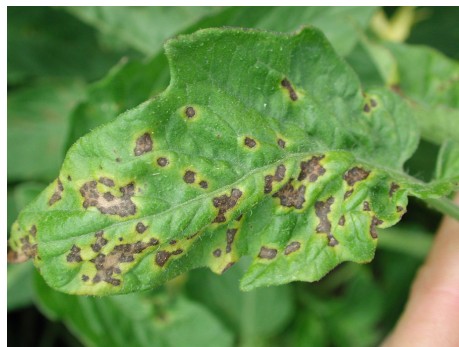

Figure 1: Example of source difference in AgriPath-LF16. **Left**: Lab-sourced image of Tomato with Bacterial Spot. **Right**: Field-sourced image of the same disease, illustrating background clutter and lighting variation.

The full dataset exhibits class imbalance and uneven domain representation, with some crop–disease pairs dominated by a single acquisition source. This imbalance complicates controlled evaluation of robustness under domain shift.

To mitigate imbalance, a balanced subset, AgriPath-LF16-30k, was constructed via class-preserving down-sampling. The subset contains 28,482 images, retains all 65 crop–disease pairs, and follows the original dataset's 80/10/10 train–validation–test split with class-stratified sampling to ensure consistent coverage across all crop–disease pairs. Balancing this subset prioritised equitable lab–field representation within each class where data permitted. Full downsampling rules are provided in Appendix A.2.

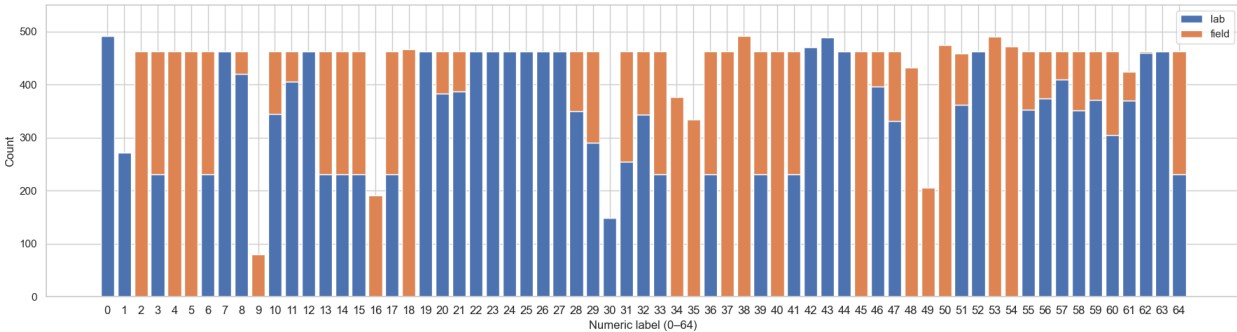

Figure 2: Class and source distribution of AgriPath-LF16-30k across 65 crop-disease pairs. Blue refers to lab-based samples and orange refers to field-based samples.

Figure 2 illustrates the distribution of AgriPath-LF16-30k across 65 crop–disease pairs, revealing a substantially flattened class profile. Although minor imbalance persists due to data availability, the subset avoids the severe long-tailed skew typical of agricultural datasets, enabling more reliable cross-class and cross-domain evaluation. The figure also shows that some classes remain dominated by lab imagery due to limited field data. This reflects practical constraints in real-world data collection rather than a deliberate design choice. Retaining this imbalance allows robustness to acquisition conditions to be evaluated under realistic deployment scenarios.

To ensure dataset integrity across aggregated sources, we conducted a post-hoc duplicate analysis using perceptual image hashing[1]. This identified a small degree of cross-split overlap (less than 6.5% of instances), which was remedied.

AgriPath-LF16-30k provides a domain-aware benchmark for evaluating performance, robustness, and generalisation across CNNs, and contrastive and generative VLMs under both controlled and in-the-wild conditions.

## 3.2 CNN Baseline

ResNet-50 (He et al., 2015) and ConvNeXt-Tiny (Liu et al., 2022) models, pre-trained on the ImageNet dataset, are used to create the CNN baseline. All backbone layers were initialised with ImageNet weights. Following standard transfer learning practice, the backbone was largely frozen, with fine-tuning restricted to the final high-level feature stage and a newly initialised classification head.

To identify an effective configuration, a small grid of nine experiments was conducted (see Appendix C for sweep details and results). This design provides a consistent set of CNN baselines for assessing in-domain performance as well as robustness under both moderate and extreme forms of distributional shift. Of the 9 experiments, the experiment with the lowest validation loss was selected for analysis. The CNNs were trained under three regimes: (a) using the full training set containing both lab and field images, (b) using only lab images, and (c) using only field images.

---

[1] https://github.com/JohannesBuchner/imagehash

### 3.3 Contrastive VLM Experiments

Contrastive VLMs employ Vision Transformer (ViT) backbones pretrained via image–text alignment objectives and perform inference over a fixed label space. Two models were evaluated: Google's SigLIP base model ($\approx$203M parameters) (Zhai et al., 2023) and OpenAI's CLIP ViT-L/14 ($\approx$427M parameters) (Radford et al., 2021). These contrastive models were evaluated in two settings: 1) *Zero-shot classification* and 2) *linear probing.*

In zero-shot classification, images and textual class descriptions are embedded into a shared representation space, with labels assigned via cosine similarity. For each class, descriptive templates were constructed from crop and disease metadata (see Appendix D.1 for the zero-shot templates), and similarity scores were averaged across a small template ensemble.

For linear probing, we train a linear classifier using frozen image embeddings, updating only classifier parameters while keeping the pretrained vision encoder fixed. This is done for the same three regimes as the CNN. For further details on the implementation of linear probing, see Appendix D.2.

### 3.4 Generative VLM Experiments

In contrast to encoder-only contrastive models, generative VLMs produce free-text outputs conditioned on visual input. Three generative VLMs of varying scale are evaluated: SmolVLM 500M Instruct (Marafioti et al., 2025), Qwen2.5-VL 3B Instruct, and Qwen2.5-VL 7B Instruct (Bai et al., 2025).

To isolate inherent model capabilities for the task, zero-shot performance was evaluated prior to adaptation using three prompting strategies of increasing output constraint: (1) a simple instruction (Pure), (2) a context-augmented instruction providing the full set of crop–disease pairs (Context), and (3) a multiple-choice format (MCQ) requiring selection from four candidate classes. However, zero-shot performance is confounded by low PSR, making it difficult to disentangle instruction-following failures from visual recognition capability. Therefore, a frozen-vision (FV) condition was included, where the vision encoder remained fixed and LoRA was applied only to the language components.

To adapt the selected VLMs to the task, Low-Rank Adaptation (LoRA) (Hu et al., 2022) was applied to the models, with all experiments using a multimodal conversational format (Appendix E.2). A Bayesian optimisation sweep tuned learning rate, weight decay, and LoRA rank $r$ (see Appendix E.3 for sweep configurations). A scaling factor of $\alpha = 2r$ was adopted after preliminary experiments showed consistently lower validation loss compared to $\alpha = r$. This is done for the same three regimes as the aforementioned models.

## 4 Evaluation & Analysis

### 4.1 Evaluation Setup

Three model families are evaluated under a unified protocol: CNNs, contrastive VLMs, and generative VLMs. CNNs and contrastive models produce closed-set predictions, whereas generative VLMs output free-text responses that must be mapped to valid class labels. Performance is evaluated using macro-averaged Precision, Recall, and F1-score across the 65 classes, with F1 reported as the primary metric. Generalisation is assessed by evaluating performance separately on lab, field, and combined test sets, enabling analysis under domain shift. Results for overall performance are reported in Table 1, while generalisation experiments can be found in Table 2.

To ensure fair comparison, generative outputs that cannot be parsed into valid class labels are treated as incorrect predictions via assignment to a dedicated `false_parse` class. PSR is reported separately to distinguish visual misclassification from instruction-related failures.

|  | PSR (%) | Macro-F1 | Macro-Precision | Macro-Recall |
|---|---|---|---|---|
| Random | – | **1.9** | 1.9 | 1.9 |
| Majority | – | 0.07 | 0.03 | 1.5 |
| ResNet-50 | – | 91.0 | 92.2 | 90.7 |
| ConvNeXt-Tiny | – | **92.9** | 93.2 | 93.0 |
| SigLIP-ZS | – | 0.2 | 0.2 | 0.7 |
| CLIP/L/14-ZS | – | **14.3** | 19.8 | 19.0 |
| SigLIP | – | 90.0 | 90.4 | 90.0 |
| CLIP/L/14 | – | **91.1** | 91.6 | 91.0 |
| SmolVLM-500M-ZS-Context | 23.8 | 0.2 | 0.6 | 0.4 |
| Qwen2.5-VL-3B-ZS-MCQ | 21.3 | 24.6 | 66.2 | 17.1 |
| Qwen2.5-VL-7B-ZS-MCQ | 94.9 | **66.9** | 71.6 | 66.6 |
| SmolVLM-500M-FV | 99.9 | 75.3 | 76.3 | 75.9 |
| Qwen2.5-VL-3B-FV | 99.9 | 87.0 | 87.7 | 86.9 |
| Qwen2.5-VL-7B-FV | 100.0 | **89.9** | 90.5 | 90.1 |
| SmolVLM-500M | 100.0 | 87.9 | 89.8 | 87.5 |
| Qwen2.5-VL-3B | 99.9 | 88.8 | 89.3 | 88.8 |
| Qwen2.5-VL-7B | 99.8 | **90.5** | 90.8 | 90.5 |

Table 1: **Overall performance on the combined test set.** Macro-F1, precision, recall, and Parse Success Rate (PSR) are reported in percentage scale (0–100). PSR is applicable only to generative VLMs. Bold values denote the strongest F1-score within each experimental group.

## 4.2 CNN-Based Models under Domain Shift

To calibrate task difficulty, two naïve reference classifiers were evaluated. A Random classifier achieved an F1-score of 1.9, consistent with chance performance, while a Majority-class baseline achieved 1.5 macro-recall, indicating the absence of a dominant class and confirming the non-triviality of the task.

CNN baselines (ResNet-50 and ConvNeXt-Tiny), pretrained on ImageNet and fine-tuned for crop disease classification, represent conventional supervised approaches (Islam et al., 2023; K.M. et al., 2023). When trained and evaluated on the combined lab and field dataset, CNN models achieve F1-scores above 91, demonstrating strong performance when training and test conditions are aligned, with ConvNeXt-Tiny achieving the strongest combined and field performance within this family. Both models attain lab-only F1-scores above 97, indicating particularly strong performance in controlled imaging environments.

However, performance is highly sensitive to changes in data distribution. As shown in Table 2, CNN performance decreases substantially under domain shift, with F1-scores dropping from ≈97 (Lab) to ≈70–75 (Field) under full-dataset training (≈25.6% relative reduction). Under more extreme mismatch, CNNs trained only on lab images achieve lab F1-scores above 96 but degrade to field F1-scores below 7, indicating near-complete failure to transfer to in-the-wild conditions (≈94.5% collapse). The converse pattern is also observed: models trained only on field data achieve strong field performance (up to 78.8) but perform poorly on lab images (≈13) relative to other architectures.

Taken together, these results indicate that CNN-based classifiers learn representations that are highly sensitive to training domain composition. While ConvNeXt-Tiny improves cross-domain performance relative to ResNet-50, both architectures exhibit similar failure under extreme domain mismatch. This behaviour suggests that CNNs are well suited to controlled imaging pipelines, but their reliability degrades substantially when applied across heterogeneous acquisition settings.

### 4.3 Contrastive Vision-Language Models

Contrastive VLMs are evaluated in both zero-shot and linear-probe settings under the same full, lab-only, and field-only regimes used for CNNs.

Zero-shot inference performs poorly (F1 $\leq$ 15) as the cosine similarity between image embeddings and templated class prompts does not reliably separate the 65 crop–disease categories. This indicates that web-scale contrastive alignment alone does not encode the fine-grained distinctions required, and that task-specific adaptation is necessary.

Linear probing substantially improves performance, showing that pretrained contrastive encoders already contain disease-relevant features. Under full-dataset training, performance increases with model capacity; contrastive models achieve performance comparable to CNNs on lab images while maintaining competitive cross-domain performance on field data, suggesting that their pretrained representations are less tightly coupled to controlled imaging conditions.

Under domain-exclusive training, contrastive VLMs degrade on the opposite domain: field-trained models transfer to lab settings better than the inverse. This suggests that lab-only training encourages reliance on clean, low-variance cues that do not transfer to real-world settings, whereas field-only training yields classifiers that tolerate cleaner lab imagery. Compared to CNNs, contrastive VLMs exhibit similar but slightly more stable cross-domain behaviour, though substantial degradation persists under extreme lab-to-field mismatch.

Contrastive models operate over a fixed label space with deterministic inference, eliminating parse failures and interface-induced errors. Their closed-set formulation constrains output flexibility but provides predictable and structurally reliable predictions. Overall, contrastive VLMs offer a domain-tolerant alternative to conventional CNNs while maintaining architectural simplicity and minimal task-specific adaptation.

### 4.4 Generative Vision-Language Models

Generative VLMs extend contrastive alignment by coupling pretrained vision encoders with large language models capable of free-text reasoning and structured output generation. Their behaviour is examined under increasing levels of task adaptation, from zero-shot prompting to supervised LoRA fine-tuning.

In the zero-shot setting, performance is highly sensitive to prompt formulation. Simple instruction prompts yield uniformly low F1-scores ($\leq$ 4), whereas structurally constrained multiple-choice prompts substantially improve performance for larger models (see Appendix Table 11 for full results). This contrast indicates that task-relevant visual knowledge is present in pretrained representations, but zero-shot performance is limited by the model's ability to reliably map visual evidence into the required structured output format. Unlike contrastive models, inference quality is therefore jointly determined by representation and interface design.

The frozen-vision (FV) condition isolates the contribution of pretrained visual features by applying LoRA only to language components. Under this constraint, both Qwen variants retain strong classification performance (F1 $\geq$ 87), implying that substantial disease-relevant signal is already encoded in the vision backbone. SmolVLM exhibits greater degradation, suggesting that model capacity influences how effectively frozen visual representations can be exploited.

Under full LoRA fine-tuning on the combined dataset, generative VLMs achieve F1-scores $\geq$ 87.5, competitive with CNNs. While CNNs achieve higher lab accuracy, generative VLMs show more balanced performance across domains. On average, generative models experience a smaller relative drop under domain shift ($\approx$ 20.2%) compared to CNNs ($\approx$ 25.6%), with robustness improving with model scale.

Cross-domain generalisation is evaluated via domain-exclusive fine-tuning. Lab-only training leads to substantial degradation when evaluated on field data, with varying severity across models. Qwen-based models exhibit a smaller relative drop ($\approx$ 73.5%) compared to the CNN collapse ($\approx$ 94.5%), retaining a meaningful fraction of their performance under severe mismatch. SmolVLM shows a larger decline ($\approx$ 80.6%), suggesting that model capacity influences the retention of transferable visual features under extreme domain shift.

As observed with contrastive models, transfer asymmetry emerges, but in the opposite direction: lab-to-field degradation is consistently larger than field-to-lab. This indicates that models trained on field data

learn representations that better tolerate cleaner laboratory conditions, whereas models trained only on lab imagery struggle to generalise to the higher variability of field environments.

Overall, generative VLMs exhibit strong pretrained visual representations that can be unlocked with modest supervision, though their effectiveness depends on training domain and adaptation strategy. Their generative interface introduces additional complexity, but also supports improved robustness under domain shift, reflecting a trade-off between interface simplicity and representational generality.

|  | PSR (%) | F1(Combined) | F1(Lab) | F1(Field) |
|---|---|---|---|---|
| *Trained on full dataset* | | | | |
| ResNet-50 | – | 91.0 | **97.2** | 69.9 |
| ConvNeXt-Tiny | – | **92.9** | 97.1 | 74.6 |
| SigLIP | – | 90.0 | 93.1 | 72.9 |
| CLIP/L/14 | – | 91.1 | 95.6 | 74.5 |
| SmolVLM-500M | 100.0 | 87.9 | 92.9 | 68.7 |
| Qwen2.5-VL-3B | 99.9 | 88.8 | 92.0 | 74.5 |
| Qwen2.5-VL-7B | 99.8 | 90.5 | 92.8 | **78.4** |
| *Trained on lab-only dataset* | | | | |
| ResNet-50 | – | 56.4 | 96.7 | 4.3 |
| ConvNeXt-Tiny | – | 56.9 | **97.5** | 6.3 |
| SigLIP | – | 56.4 | 94.9 | 13.5 |
| CLIP/L/14 | – | 57.1 | 95.8 | 11.0 |
| SmolVLM-500M | 99.9 | **59.3** | 94.5 | 18.3 |
| Qwen2.5-VL-3B | 94.1 | 55.3 | 88.8 | 23.8 |
| Qwen2.5-VL-7B | 100.0 | 58.6 | 94.1 | **24.6** |
| *Trained on field-only dataset* | | | | |
| ResNet-50 | – | 37.1 | 13.4 | 73.2 |
| ConvNeXt-Tiny | – | 40.1 | 12.8 | **78.8** |
| SigLIP | – | 41.1 | 19.8 | 75.6 |
| CLIP/L/14 | – | 40.9 | 17.4 | 77.0 |
| SmolVLM-500M | 99.9 | 34.8 | 10.6 | 67.3 |
| Qwen2.5-VL-3B | 99.1 | 39.0 | 15.7 | 73.7 |
| Qwen2.5-VL-7B | 91.8 | **42.9** | **20.7** | 73.8 |

Table 2: **Generalisation performance across domain-specific training regimes.** Macro-F1 scores (percentage scale) are reported for combined, Lab-only, and Field-only test sets. PSR is shown for generative models. Bold values indicate the strongest F1 within each test domain.

### 4.5 Fine-Grained Analysis under Domain-Constrained Evaluation

To analyse class-level behaviour while controlling for total data per class, crop-disease pairs were grouped by training domain composition (balanced vs. lab-dominant). To ensure stable estimates, only classes with sufficient field support in the test set ($\geq 11$ samples; max $=23$) were retained, yielding 12 balanced and 8 lab-dominant classes for analysis.

This filtering reveals a key dataset constraint: reliable per-class field evaluation is concentrated in balanced classes, while extremely lab-dominant classes are largely unevaluable due to insufficient field support. This indicates that class-level field performance is inherently limited by domain-specific data availability, rather than uniformly supported across all classes.

On balanced classes, where domain imbalance is largely removed, all models achieve similar in-domain performance (mean lab F1 $\approx 0.93$–$0.94$), but differ in cross-domain generalisation: CNN (field F1 0.820, gap 0.118), contrastive VLM (0.792, 0.147), and generative VLM (0.824, 0.106). This shows that architectural differences persist even under controlled conditions, with generative VLMs exhibiting the lowest domain sensitivity, albeit minimal.

When moving to lab-dominant classes, degradation increases for all models, but substantially more for CNNs. The domain gap increases by +0.146 for CNNs, compared to +0.085 for contrastive VLMs and +0.076 for generative VLMs, with corresponding reductions in field F1 (0.721 for CNN, 0.722 for contrastive, 0.766 for generative). This indicates that CNNs are more sensitive to reduced field exposure, while VLMs exhibit more stable cross-domain behaviour as domain imbalance increases.

Together the results show that domain composition acts as a controlling factor for generalisation. While all architectures benefit from balanced domain exposure, CNN performance degrades more sharply as field representation decreases, whereas VLMs retain more stable performance under domain-constrained conditions.

### 4.6 Failure Analysis

This section analyses the different failure modes by providing specific examples of these failure modes that are characteristic of each model type.

#### 4.6.1 CNN Failure Modes

To analyse CNN failure modes, the confusion matrix of the ResNet50 model trained on combined lab and field data was examined (Appendix F.1). The matrix revealed that `corn_common_rust` (Class 13) exhibited the highest error rate, primarily being misclassified as `corn_phaeosphaeria_leaf_spot` (Class 18). Figure 3 illustrates a representative case of this inter-class leakage.

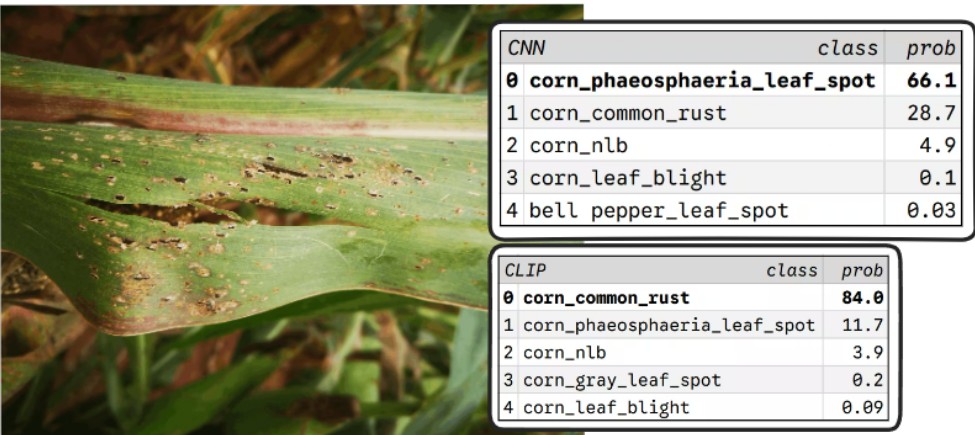

Figure 3: A corn crop with common rust in the field with probabilities for a CNN and CLIP prediction.

The input image features a damaged leaf where necrotic lesions and perforations dominate the local texture, alongside darker rust-coloured pustules distributed across the leaf surface. The CNN's preference for leaf spot ($\approx 66\%$) may stem from an over-reliance on shape-dominant local features under distributional noise, reflecting an inductive bias toward lesion geometry and texture patterns. The VLM, however, aligns the image with higher-level semantic concepts such as rust-like visual cues, resulting in correct identification of the rust disease with high confidence ($\approx 84\%$). Despite this, the non-negligible probability assigned to the correct class ($\approx 28\%$) indicates that the CNN maintains partial recognition of the underlying pathology.

#### 4.6.2 Contrastive VLM Failure Modes

Similar to the CNN, the confusion matrix of the CLIP model was consulted (Appendix F.2), and an example where the CLIP model exhibits pronounced uncertainty between two closely related disease classes is examined. Figure 4 shows a potato leaf affected by blight-like symptoms, for which the model assigns comparable probability to `potato_late_blight` (45%) and `potato_early_blight` (42%). Generally, both diseases present overlapping visual phenotypic characteristics, including irregular necrotic regions and diffuse chlorosis, making disambiguation challenging.

Figure 4: A potato crop with an ambiguous blight disease in the field with an uncertain CLIP prediction.

Rather than a spurious misclassification, this example reflects difficulty in resolving fine-grained class boundaries under substantial visual overlap. The model's probability distribution reveals semantic proximity between the two classes in the embedding space, indicating that the visual evidence does not strongly favour a single diagnosis. Although this uncertainty is preferable to overconfident error, it remains problematic in high-stakes agricultural contexts, where early and late blight require different interventions. This illustrates a limitation of contrastive VLMs: strong global semantic alignment does not guarantee reliable separation of subtle intra-class distinctions without additional contextual or temporal cues.

### 4.6.3 Generative VLM Failure Modes

Unconstrained text generation introduces failure modes not present in discriminative models. In this setting, these failures are not arbitrary but fall into a small number of structured categories arising from the interaction between free-text generation and a constrained crop–disease label space. Specifically, four recurring failure types are observed:

First, models introduce non-existent modifiers or entities (e.g., `icelandic raspberry`, `icicle apple`), producing crop–disease combinations outside the dataset ontology. Additionally, unsupported crop substitutions appear (e.g., `banana`, `sage`, `hazel`), despite their absence from the training distribution. Second, invalid attribution compositions could occur where valid crop and disease tokens are combined into non-existent classes. Third, other failures stem from semantic vagueness rather than outright fabrication. For instance, models can output underspecified classes (e.g., `leaf_spot`, `tomato_spot`) that correspond to multiple dataset classes and cannot be resolved unambiguously. Finally, 39.7% of all false parses returned blank raw outputs, which are treated as silent failures and would not occur in a fixed label space.

These findings indicate that although generative VLMs offer expressive and flexible predictions, they introduce non-trivial risks of hallucination and silent failure in high-stakes agricultural deployment, exacerbated under domain shift, where uncertainty in visual recognition increases the likelihood of underspecified, invalid, or empty generations.

### 4.7 Summary of Findings

Results show that crop disease detection depends heavily on the deployment context. CNNs achieve the strongest performance on lab-based data, but experience the steepest degradation under domain shift. Consistent with prior findings that CNN representations are tightly coupled to their training distribution, their reliance on local texture and shape cues makes them sensitive to changes in environmental context, resulting in reduced aggregate performance and systematic class-level confusions under field conditions.

Contrastive VLMs demonstrate improved robustness to domain shift relative to CNNs. Their errors tend to cluster around visually or semantically adjacent classes, indicating uncertainty rather than complete failure. This suggests that contrastive pretraining encourages more transferable visual representations, while still limited by visually subtle, fine-grained disease distinctions.

Generative VLMs are the most robust to distributional shift among the three model types, despite their robustness, their open-ended generation objective leads to non-trivial rates of invalid or hallucinated outputs that are critical in safety-sensitive agricultural contexts.

Taken together, these findings indicate that no single model family is universally optimal. Instead, deployment requirements should guide model selection: CNNs are preferred when high lab-based accuracy is the primary objective; contrastive VLMs offer a more robust alternative for fixed sets of crops under moderate domain shift; and generative VLMs are most suitable when flexibility and extended capabilities are required.

## 5   Conclusions & Outlook

This work presents a systematic comparison of model architectures across realistic agricultural domains and introduces a balanced benchmark for crop disease evaluation. The results show that architectural suitability depends on deployment context rather than peak accuracy.

### 5.1   Limitations & Future Work

Despite providing a controlled benchmark for cross-domain evaluation, several limitations remain. First, although AgriPath-LF16 reduces extreme imbalance, field-sourced samples remain comparatively underrepresented for certain crop–disease pairs. This limits the ability to fully characterise robustness under real-world variability, as class-level field performance becomes less reliable for underrepresented classes and aggregate metrics may be biased toward lab-dominant distributions. Second, as the dataset aggregates multiple public sources, residual redundancy or near-duplicate samples may exist despite standard preprocessing. The dataset also lacks contextual or temporal information that represents disease progression, which is dynamic and ambiguous in isolation. Additionally, multimodal agricultural metadata such as region, climate, soil characteristics, or seasonality were not incorporated. Computational constraints limited exploration of larger architectures, broader hyperparameter sweeps, and extended ablations.

Future work should expand data collection to include more field samples as well as fine-grained information such as region, climate, soil characteristics, and season to reduce domain imbalance and increase environmental coverage. Additionally, integrating contextual metadata into generative VLM pipelines would better capitalise on the multimodal capabilities of VLMs (Yao et al., 2023). Temporal modelling of disease progression using longitudinal imagery is another avenue to explore. Moreover, reasoning-aware fine-tuning that promotes localisation of symptomatic regions and suppresses background cues would allow further exploration of different architectures when translating across domains with varying levels of environmental visual noise (Bansal et al., 2025). Finally, real-world, farmer-facing evaluations would clarify usability, trust calibration, and practical performance beyond benchmarks.

### 5.2   Statement of Broader Impact

This study supports the development of robust, context-aware disease detection systems to enhance agricultural productivity and food security. Early, accurate identification of plant diseases minimises yield loss, enables timely intervention, and reduces unnecessary pesticide use. Consequently, architectures that maintain robustness under domain shift are critical for real-world agricultural settings.

Generative VLMs offer flexible diagnostic potential by integrating contextual metadata and providing structured reasoning, improving accessibility for smallholder farmers and facilitating integration into precision agriculture workflows. However, these may produce hallucinated outputs that could lead to incorrect interventions; mitigation requires structured constraints, grounding, and human oversight. Furthermore, dataset imbalance and limited field data can bias models toward controlled conditions, requiring continued domain-diverse collection. Regional bias remains a risk due to geographic variation in disease appearance; mitigation requires local data collection, fine-tuning, and transparent deployment limits. Environmental and computational costs also remain a concern. Although parameter-efficient tuning reduces overhead, architectural choices must balance sustainability with accuracy and robustness. Ultimately, careful design, transparent reporting, and responsible deployment are essential for equitable agricultural benefit.

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

# A  Dataset Composition

## A.1  Source Datasets

| Dataset Name | Source | Crop | # Diseases | # Samples | Citation/Link |
|---|---|---|---|---|---|
| New Bangladeshi Crop Disease | Lab | Rice | 4 | 4,078 | Moin (2023) |
| PlantVillage | Lab | Apple | 4 | 4,645 | G. & J. (2019) |
| | | Blueberry | 1 | 1,502 | |
| | | Cherry | 2 | 2,502 | |
| | | Corn | 4 | 4,345 | |
| | | Grape | 4 | 4,639 | |
| | | Orange | 1 | 5,507 | |
| | | Peach | 2 | 3,297 | |
| | | Bell Pepper | 2 | 2,478 | |
| | | Potato | 3 | 3,000 | |
| | | Raspberry | 1 | 1,000 | |
| | | Soybean | 1 | 5,090 | |
| | | Squash | 1 | 1,835 | |
| | | Strawberry | 2 | 2,109 | |
| | | Tomato | 10 | 18,835 | |
| PlantDoc | Field | Apple | 3 | 287 | Jain & Kayal (2024) |
| | | Bell Pepper | 2 | 125 | |
| | | Blueberry | 1 | 117 | |
| | | Cherry | 1 | 57 | |
| | | Corn | 3 | 378 | |
| | | Grape | 2 | 154 | |
| | | Peach | 1 | 112 | |
| | | Potato | 2 | 379 | |
| | | Raspberry | 1 | 119 | |
| | | Soybean | 1 | 65 | |
| | | Squash | 1 | 130 | |
| | | Strawberry | 2 | 96 | |
| | | Tomato | 9 | 903 | |
| Apple Dataset 2021 | Field | Apple | 5 | 15,675 | FGVC (2021) |
| Roboflow Rice Disease Dataset | Field | Rice | 6 | 3,733 | Jain & Kayal (2024) |
| Paddy Doctor | Field | Rice | 10 | 10,407 | Paddy Doctor (2023) |
| Rice Leaf Disease | Field | Rice | 4 | 5,932 | Sethy (2020) |
| CD&S Dataset | Field | Corn | 1 | 523 | Ahmad (2021) |
| Disease of Maize in the Field | Field | Corn | 5 | 2,716 | UoPretoria (2022) |
| CornNLB | Field | Corn | 1 | 1,787 | Brahimi et al. (2018) |
| Zeytin Olive Leaf Disease | Lab | Olive | 3 | 5,011 | Uguz & Uysal (2021) |
| Strawberry Disease Detection | Field | Strawberry | 5 | 2,268 | Afzaal et al. (2021) |

Table 3: Datasets used to compile AgriPath-LF16 along with links to the download source

### A.2 Downsampling Strategy

Below is the downsampling logic used to create AgriPath-LF16-30k. The balanced subset of AgriPath-LF16 used in this paper:

- **Case 1 (Single Source):** If samples existed for only one source (lab or field), all samples were kept if the count was close to the downsample target (within ±10), or downsampled to match the target if the count exceeded it by more than 20.

- **Case 2L (Lab less than target, field more):** If the total lab and field samples were less than the target, all samples were kept. Otherwise, field samples were downsampled to the required count, and all lab samples were included.

- **Case 2F (Field less than target, lab more):** Similar to Case 2L, if the total samples were less than the target, all samples were kept. Otherwise, lab samples were downsampled to the required count, and all field samples were included.

- **Case 3 (Both less than target):** For under-represented classes, the crop-disease pair was flagged for review, and all existing samples were kept and added to the dataset for inspection.

- **Case 4 (Both meet or exceed target):** Both lab and field samples were downsampled to equal amounts to achieve a 50/50 split and match the target sample size.

### A.3 Dataset Structure

Each sample in AgriPath-LF16 includes the following structured metadata:

- `image`: The image of the crop.

- `crop`: The specific crop (leaf) represented in the image.

- `disease`: The disease present (if any) in the image.

- `source`: Indicates whether the image was taken in a lab or in the field.

- `crop_disease_label`: A combined label representing crop-disease pairs.

- `numeric_label`: The numeric encoding (0-64) for each crop-disease pair.

## B    Training Efficiency Across Architectures

|  | Total Params | Trainable Params | Ratio (%) |
|---|---|---|---|
| *Convolutional Neural Networks* | | | |
| ResNet-50 | 23.6M | 15.1M | 63.98 |
| ConvNeXt-Tiny | 27.9M | 14.3M | 51.25 |
| *Contrastive Vision-Language Models* | | | |
| SigLIP | 203M | 50k | 0.024 |
| CLIP-L/14 | 427M | 50k | 0.012 |
| *Frozen-Vision Generative Vision-Language Models* | | | |
| Qwen2.5-VL-3B-FV | 4.0B | 240M | 6.00 |
| Qwen2.5-VL-7B-FV | 8.6B | 323M | 3.75 |
| SmolVLM-500M-FV | 584M | 76.5M | 13.11 |
| *Generative Vision-Language Models* | | | |
| Qwen2.5-VL-3B-LoRA/Lab/Field | 4.0B | 164M | 4.19 |
| Qwen2.5-VL-7B-LoRA/Lab/Field | 8.7B | 413M | 4.74 |
| SmolVLM-500M-LoRA/Lab/Field | 596M | 88M | 14.83 |

Table 4: **Model size and trainable parameter ratios.** Total and trainable parameters for each architecture and adaptation regime.

## C    Additional CNN Information

All CNN experiments were implemented using PyTorch Lightning. A coarse grid search over batch size and learning rate was conducted for both the ResNet-50 and ConvNeXt-Tiny model across all experiments. The grid search includes three batch sizes (16, 32, and 64) and three learning rates (1e-4, 2e-4, 5e-4), resulting in a total of 9 experiments. The experiment mentioned in the paper is in bold.

### C.1    ResNet-50 Experiments

**Trained on the full dataset**

| Batch Size | Learning Rate | F1(Combined) | F1(Lab) | F1(Field) |
|---|---|---|---|---|
| **16** | **1e-4** | **91.0** | **97.2** | **69.9** |
| 16 | 2e-4 | 89.3 | 95.5 | 66.1 |
| 16 | 5e-4 | 88.7 | 94.8 | 69.3 |
| 32 | 1e-4 | 90.7 | 94.0 | 70.0 |
| 32 | 2e-4 | 90.0 | 93.7 | 69.3 |
| 32 | 5e-4 | 88.8 | 91.4 | 63.8 |
| 64 | 1e-4 | 89.8 | 94.8 | 65.5 |
| 64 | 2e-4 | 89.2 | 94.9 | 64.2 |
| 64 | 5e-4 | 87.3 | 91.3 | 67.8 |

Table 5: The F1-Scores of the ResNet-50 experiments trained on the full dataset (lab and field)

**Trained on the lab-only subset of the dataset**

| Batch Size | Learning Rate | F1(Combined) | F1(Lab) | F1(Field) |
|---|---|---|---|---|
| **16** | **1e-4** | **56.4** | **96.7** | **4.3** |
| 16 | 2e-4 | 55.4 | 94.4 | 4.6 |
| 16 | 5e-4 | 55.7 | 96.8 | 3.6 |
| 32 | 1e-4 | 56.0 | 97.0 | 4.0 |
| 32 | 2e-4 | 55.6 | 95.4 | 4.8 |
| 32 | 5e-4 | 56.3 | 96.3 | 3.5 |
| 64 | 1e-4 | 55.8 | 95.8 | 4.5 |
| 64 | 2e-4 | 56.2 | 96.5 | 4.2 |
| 64 | 5e-4 | 55.3 | 95.4 | 4.2 |

Table 6: The F1-Scores of the ResNet-50 experiments trained on the lab-only subset of the full dataset

**Trained on the field-only subset of the dataset**

| Batch Size | Learning Rate | F1 (Combined) | F1(Lab) | F1(Field) |
|---|---|---|---|---|
| **16** | **1e-4** | **37.1** | **13.4** | **73.2** |
| 16 | 2e-4 | 35.4 | 11.0 | 69.0 |
| 16 | 5e-4 | 36.7 | 11.8 | 69.5 |
| 32 | 1e-4 | 40.1 | 15.3 | 76.7 |
| 32 | 2e-4 | 33.8 | 8.4 | 68.5 |
| 32 | 5e-4 | 34.4 | 11.4 | 68.3 |
| 64 | 1e-4 | 37.0 | 12.8 | 72.9 |
| 64 | 2e-4 | 37.5 | 13.0 | 70.4 |
| 64 | 5e-4 | 36.5 | 11.4 | 71.0 |

Table 7: The F1-Scores of the ResNet-50 experiments trained on the field-only subset of the full dataset

## C.2 ConvNeXt-Tiny Experiments

**Trained on the full dataset**

| Batch Size | Learning Rate | F1(Combined) | F1(Lab) | F1(Field) |
|---|---|---|---|---|
| **16** | **1e-4** | **92.9** | **97.1** | **74.6** |
| 16 | 2e-4 | 92.1 | 96.5 | 71.9 |
| 16 | 5e-4 | 92.6 | 97.0 | 73.6 |
| 32 | 1e-4 | 93.5 | 97.3 | 76.7 |
| 32 | 2e-4 | 92.4 | 97.3 | 70.5 |
| 32 | 5e-4 | 91.6 | 95.2 | 69.2 |
| 64 | 1e-4 | 92.7 | 95.1 | 73.8 |
| 64 | 2e-4 | 93.3 | 95.3 | 76.4 |
| 64 | 5e-4 | 91.4 | 94.4 | 69.6 |

Table 8: The F1-Scores of the ConvNeXt-Tiny experiments trained on the full dataset (lab and field)

**Trained on the lab-only subset of the dataset**

| Batch Size | Learning Rate | F1(Combined) | F1(Lab) | F1(Field) |
|---|---|---|---|---|
| **16** | **1e-4** | **56.9** | **97.5** | **6.3** |
| 16 | 2e-4 | 56.7 | 97.1 | 6.3 |
| 16 | 5e-4 | 56.2 | 96.9 | 5.8 |
| 32 | 1e-4 | 57.9 | 98.1 | 6.5 |
| 32 | 2e-4 | 56.5 | 96.3 | 7.6 |
| 32 | 5e-4 | 56.0 | 96.2 | 6.0 |
| 64 | 1e-4 | 57.4 | 97.2 | 6.2 |
| 64 | 2e-4 | 56.7 | 97.5 | 6.0 |
| 64 | 5e-4 | 55.5 | 95.7 | 5.5 |

Table 9: The F1-Scores of the ConvNeXt-Tiny experiments trained on the lab-only subset of the full dataset

**Trained on the field-only subset of the dataset**

| Batch Size | Learning Rate | F1(Combined) | F1(Lab) | F1(Field) |
|---|---|---|---|---|
| **16** | **1e-4** | **40.1** | **12.8** | **78.8** |
| 16 | 2e-4 | 39.2 | 13.8 | 74.4 |
| 16 | 5e-4 | 37.6 | 12.4 | 73.4 |
| 32 | 1e-4 | 40.1 | 12.0 | 79.7 |
| 32 | 2e-4 | 41.7 | 14.3 | 80.0 |
| 32 | 5e-4 | 38.8 | 13.8 | 74.9 |
| 64 | 1e-4 | 40.2 | 12.2 | 77.1 |
| 64 | 2e-4 | 40.3 | 12.7 | 79.8 |
| 64 | 5e-4 | 40.4 | 14.0 | 76.2 |

Table 10: The F1-Scores of the ConvNeXt-Tiny experiments trained on the field-only subset of the full dataset

# D   Additional Contrastive VLM Information

## D.1   Zero-Shot Templates

Zero-shot classification for contrastive VLMs was performed using prompt-ensemble prototypes constructed from crop–disease metadata. Text prompts were generated using class-specific templates and averaged in embedding space prior to similarity computation. Two template families were defined:

1. Diseased Templates:
   - `"a photo of a {crop} leaf with {disease}"`
   - `"an image of a {crop} leaf affected by {disease}"`
   - `"a close-up photo of a {crop} leaf showing {disease}"`
2. Healthy Templates:
   - `"a photo of a healthy {crop} leaf"`
   - `"an image of a healthy {crop} leaf"`
   - `"a close-up photo of a healthy {crop} leaf"`

For each of the 65 crop–disease classes, all corresponding templates were instantiated, encoded using the model's text encoder, L2-normalised, and averaged to form a class prototype vector. During inference, image embeddings were L2-normalised and cosine similarity was computed against all class prototypes. The predicted label corresponds to the maximum similarity score.

## D.2   Linear Probing Configuration

To evaluate the quality of pretrained visual representations, linear probing was performed by training a single fully-connected classifier head on frozen image embeddings.

**Backbone Handling:**

- Pretrained SigLIP and CLIP backbones were loaded via `AutoModel`.
- All backbone parameters were frozen.
- Image embeddings were extracted using `get_image_features()` when available, or via the CLS token from the vision encoder.
- Embeddings were L2-normalised before classification.

**Classifier Head:**

A single Linear layer, `Linear(feature_dim, 65)` was trained on top of frozen features.

- Only classifier parameters were updated.
- Backbone gradients were disabled.

The trained head and metadata (feature dimension, class count) were stored as a W&B artifact and reloaded during evaluation.

**Training Configuration:**

All linear probing experiments used:

- Batch size: 64
- Cross-Entropy loss
- Macro-F1 as evaluation metric
- Learning Rate selected via small manual sweep from {0.001, 0.003, 0.01}

# E    Additional Generative VLM Information

## E.1    Zero-Shot Evaluations

|  | PSR (%) | F1 | Precision | Recall |
|---|---|---|---|---|
| SmolVLM-500M-ZS-Pure | 0.2 | 0 | 0 | 0 |
| **SmolVLM-500M-ZS-Context** | **23.8** | **0.2** | **0.6** | **0.4** |
| SmolVLM-500M-ZS-MCQ | 0 | 0 | 0 | 0 |
| Qwen2.5-VL-3B-ZS-Pure | 90.7 | 4.2 | 5.8 | 6.3 |
| Qwen2.5-VL-3B-ZS-Context | 19.1 | 2.5 | 18.9 | 2.0 |
| **Qwen2.5-VL-3B-ZS-MCQ** | **21.3** | **24.6** | **66.2** | **17.1** |
| Qwen2.5-VL-7B-ZS-Pure | 69.6 | 2.2 | 3.7 | 3.2 |
| Qwen2.5-VL-7B-ZS-Context | 91.1 | 10.6 | 17.7 | 15.4 |
| **Qwen2.5-VL-7B-ZS-MCQ** | **94.9** | **66.9** | **71.6** | **66.6** |

Table 11: All ZS experiments for all Generative VLMs, reporting PSR, F1, Precision, and Recall in a range of 0-100. Experiments mentioned in the paper are in bold

## E.2    LoRA Conversational Format

```
conversation = [
    {"role": "system",
        "content": [
            {"type": "text", "text": "You are an expert pathologist and need to identify the crop and
                disease present in an image. If it is a healthy crop, classify it as healthy"}
        ]
    },
    {"role": "user",
    "content": [
            {"type": "text", "text": "Identify the crop and disease in the image."},
            {"type": "image", "image": sample['image']}
        ]
    },
    {"role": "assistant",
    "content": [
            {"type": "text", "text": f"Class: {sample['crop']}\nDisease: {sample['disease']}"}
        ]
    }
]
```

### E.3    Sweep Parameters

All sweeps below use $\alpha = 2r$ in the LoRA configurations of the sweep.

### E.3.1    Initial Search Space

| Parameters | Distribution | Range |
|---|---|---|
| Learning Rate | Uniform | $5 \times 10^{-5}$ to $2 \times 10^{-4}$ |
| LoRA Rank $r$ | Categorical | 32, 64, 128 (Qwen3) \| 64, 128 (Qwen7) |
| Weight Decay | Uniform | 0 to 0.1 |

Table 12: The initial parameters used for sweeps. Qwen3 and Qwen7 refer to the 3B and 7B variants of the Qwen2.5-VL-xB-LoRA experiments respectively.

Early sweeps indicated stable convergence in the region $r \in \{64, 128\}$, hence the shift from $\{32, 64, 128\}$ to $\{64, 128\}$ seen above in Table 12.

### E.3.2    Refined Search Space

Based on preliminary results, the LoRA rank was fixed and subsequent sweeps were restricted to learning rate and weight decay, using a reduced search range.

| Parameters | Distribution | Range |
|---|---|---|
| Learning Rate | Uniform | $5 \times 10^{-5}$ to $1.5 \times 10^{-4}$ |
| Weight Decay | Uniform | 0 to 0.1 |
| LoRA Rank $r$ | Fixed | 128 |

Table 13: The refined parameters used for sweeps after the initial sweeps showed some trends of convergence.

### E.3.3    Summary of Sweep Configurations

| Sweep Name | Regime | Search Space |
|---|---|---|
| Qwen3 | Full LoRA | Initial |
| Qwen3-LAB | Lab LoRA | Initial |
| Qwen3-FV | Frozen Vision | Refined |
| Qwen3-FIELD | Field LoRA | Refined |
| Qwen7 | Full LoRA | Initial |
| Qwen7-LAB | Lab LoRA | Initial |
| Qwen7-FV | Frozen Vision | Refined |
| Qwen7-FIELD | Field LoRA | Refined |
| Smol | Full LoRA | Refined |
| Smol-LAB | Lab LoRA | Refined |
| Smol-FV | Frozen Vision | Refined |
| Smol-FIELD | Field LoRA | Refined |

Table 14: A summary of the search spaces used for each sweep. Qwen3 relates to Qwen2.5-VL-3B, Qwen7 relates to Qwen2.5-VL-7B, and Smol relates to SmolVLM-500M. Initial relates to appendix E.3.1, and Refined relates to appendix E.3.2

### E.4    Implementation Configuration

#### E.4.1    Training Frameworks and LoRA Implementation

Generative VLM experiments were conducted using two training backends:

- Qwen2.5-VL models (3B, 7B) were fine-tuned using the UnslothAI framework via `FastVisionModel` with PEFT-based LoRA adaptation.
- SmolVLM-500M was fine-tuned using a custom PEFT pipeline built on `Idefics3ForConditionalGeneration`, as UnslothAI does not natively support this architecture.

In both cases, LoRA was applied to attention and MLP modules. For frozen-vision experiments, LoRA was restricted to language components only, leaving vision layers untouched. All LoRA runs used:

- Scaling Factor: $\alpha = 2r$
- Dropout: 0
- Bias: none

This ensures architectural parity across implementations despite backend differences.

#### E.4.2    Inference Configuration

During evaluation:

- Models were loaded using their respective backends (UnslothAI or PEFT).
- Generation was deterministic (temperature = 0).
- Image resizing was constrained to a longest edge of 512 pixels.
- No beam search or sampling was used.

This ensured that performance differences reflect representational differences rather than stochastic decoding effects.

#### E.4.3    Output Parsing and Evaluation

Model outputs were programmatically mapped to the 65 crop–disease classes. The following regular expressions were used to extract structured predictions:

- `"(?:Class|Answer|Crop):\s*(\w+(?: \w+)*)\s*[\r\n]+Disease:\s*(\w+(?:_\w+)*)"`
- `"Answer:\s*[\r\n]+(\w+(?: \w+)*)\s*[\r\n]+(\w+(?:_\w+)*)"`
- `"Disease:\s*(\w+(?:_\w+)*)\s*[\r\n]+(?:Crop|Class|Answer):\s*(\w+(?: \w+)*)"`

Parsing then extracted the exact crop and disease fields from the generated text and matched it against canonical class list. If no valid mapping is found, the prediction is assigned to the `false_parse` class.

Empty generations and malformed outputs were treated as incorrect predictions and penalised F1 computation in order to retrieve a holistic view of model capabilities.

# F   Confusion Matrix Analysis

## F.1   CNN – ResNet-50 (Batch=16, LR=1e-4)

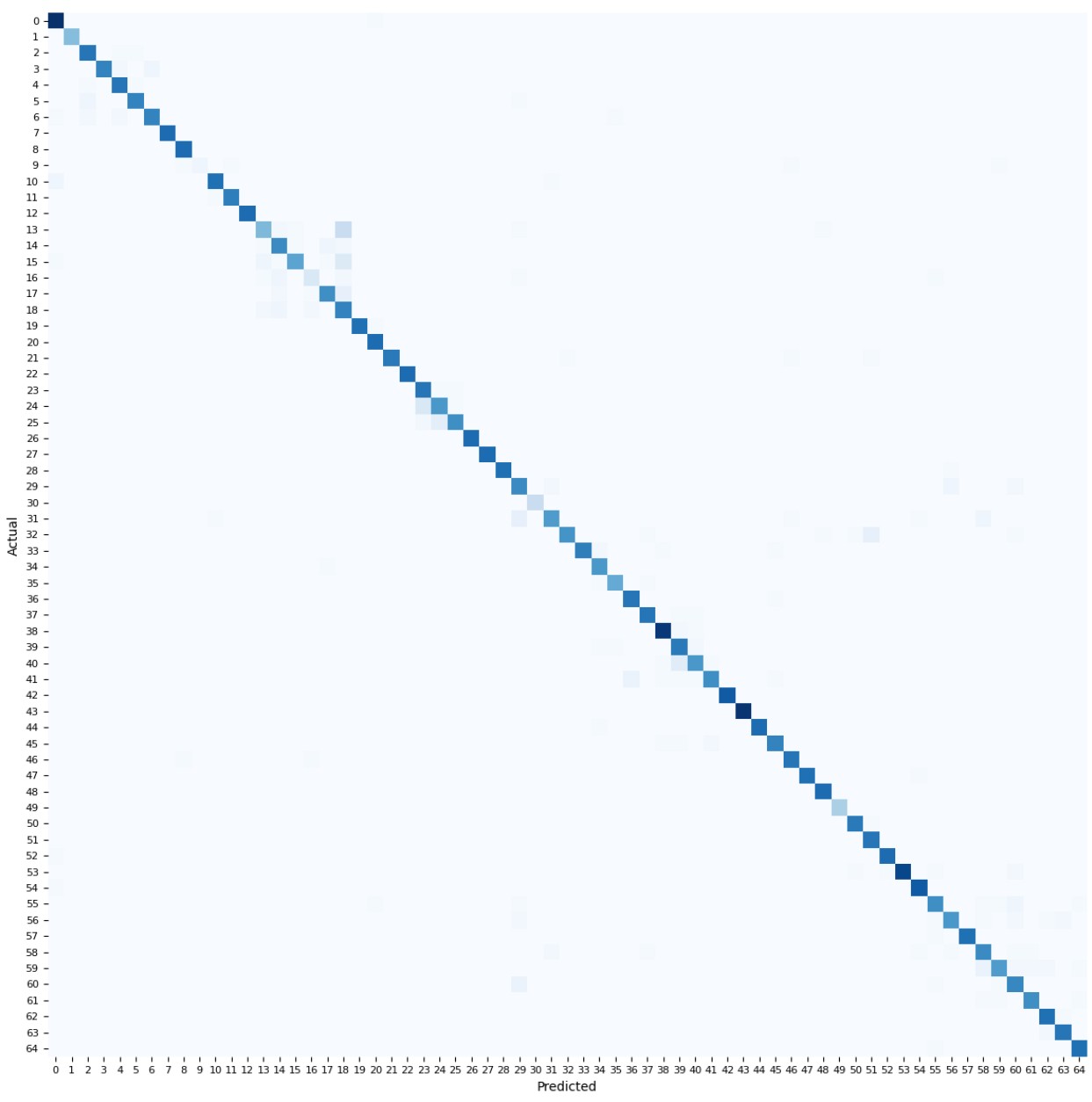

Figure 5: Top confusion pairs are discussed in Section 4.6.1. Actual labels are along the y-axis, and Predicted labels are along the x-axis.

## F.2 CLIP – CLIP/L/14

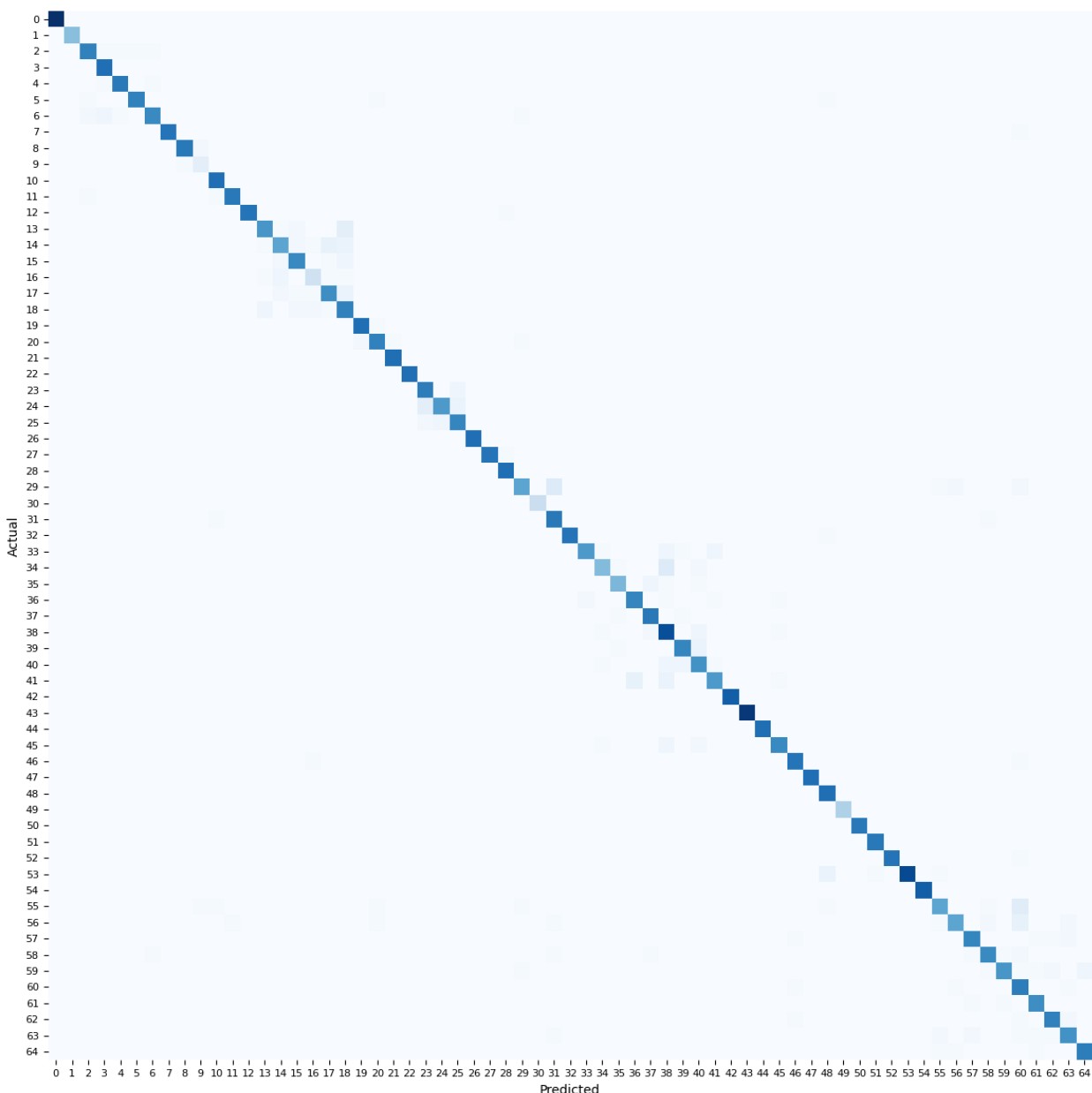

Figure 6: Top confusion pairs are discussed in Section 4.6.2. Actual labels are along the y-axis, and Predicted labels are along the x-axis.

