# OpenReview forum: "AgriPath: A Systematic Exploration of Architectural Trade- offs for Crop Disease Classification"
_TMLR — Accepted by TMLR_

### Review · Reviewer_YEJb · 2026-03-22

**Summary Of Contributions:**

This paper presents a systematic empirical comparison of CNNs, contrastive vision-language models, and generative vision-language models for crop disease classification under domain shift. It introduces a domain-aware dataset (AgriPath-LF16) with explicit lab versus field splits and evaluates models under unified training regimes. The results show clear trade-offs between accuracy, robustness, and reliability across model families, highlighting that model choice should depend on deployment context. Overall, the paper is a well-executed benchmarking study with practical relevance but limited methodological novelty.

Strengths:

- The experimental design is clean and well controlled, with consistent protocols across different model families and training regimes, which makes the comparison credible and easy to interpret.

- The dataset contribution is meaningful, especially the explicit separation between lab and field domains, which directly targets a real and underexplored issue in this application space.

- The paper provides clear and practically useful insights about trade-offs between model types, which could be helpful for deployment decisions in real-world agricultural systems.

Weaknesses:

- The main findings largely confirm well-known behavior of these model classes (e.g., CNN sensitivity to domain shift, stronger generalization of large pre-trained models, hallucination in generative models), and therefore offer limited novel insight.

- The failure modes identified for generative VLMs (such as hallucination and parsing errors) are generic to LLM-based systems and not specific to this task or domain, which reduces the depth of the analysis (we all know LLM hallucinates / generates non-parsable output regardless of the task...)

- The paper focuses on empirical comparison without providing deeper mechanistic understanding or new hypotheses about why these behaviors arise.

- The evaluation is restricted to a single task (classification), and it is unclear how broadly the conclusions generalize beyond this setting.

**Audience:**

Yes

**Audience Explanation:**

This paper would mostly interest people working on robustness and domain shift, especially in vision or multimodal models. It’s also relevant for applied ML folks in agriculture or similar real-world settings where lab vs field gaps matter. Additionally, people studying evaluation and benchmarking of foundation models might care. But overall, the audience is somewhat niche and probably not broad across all of ML.

**Broader Impact Concerns:**

Brief description of any concerns on the ethical implications of the work that would require adding a Broader Impact Statement (if one is not present) or that are not sufficiently addressed in the Broader Impact Statement section (if one is present) (max 200000 characters).

**Claims And Evidence:**

Yes

**Claims Explanation:**

Overall, yes, the claims are generally supported by the experiments within the paper.

- The comparisons are conducted under consistent settings, which supports fairness.
- The reported trends (e.g., CNN degradation, VLM robustness) are backed by quantitative results across multiple regimes.
- The conclusions are mostly aligned with the empirical evidence presented.

However:

- The claims are mostly confirmatory rather than novel, meaning they are supported, but not particularly surprising.
- Some higher-level interpretations (e.g., about robustness or deployment suitability) are reasonable but not deeply analyzed, and could benefit from stronger causal or diagnostic evidence.

**Requested Changes:**

- Clarify what is actually new in this paper beyond confirming expected trends, and better position the contribution relative to prior work on domain shift and pre-trained models.

- Provide deeper analysis on why different model families behave differently, not just reporting performance gaps.

- Make the failure analysis more specific to this task instead of focusing on generic LLM issues like hallucination and parsing errors.

- Discuss how well the conclusions generalize beyond this dataset and this specific classification setting.

- Add more discussion on dataset limitations (especially lab vs field imbalance) and how they might affect the results.

---

> ### Author Response · Authors · 2026-04-07
> **Response to Reviewer YEJb (Part 1)**
>
> We would like to thank the reviewer for their feedback. We provide a clarification to each point raised below:
>
> ---
>
> ### 1. Novelty and Better Positioning of Contributions
> >“limited methodological novelty”
> >“findings confirm well-known behaviour”
>
> We agree that individual behaviours of model classes are known in isolation. The novelty of this work lies in the controlled, unified evaluation across architectures, enabled by a benchmark with explicit lab/field separation and a metric suite that makes generative and discriminative models directly comparable.
>
> ---
>
> ### 2. Deeper Analysis
> >“lack of deeper mechanistic understanding”
> >“Provide deeper analysis on why different model families behave differently”
>
> The paper is intended as a diagnostic empirical study, not a mechanistic interpretability work. It already provides task-level diagnostic evidence through:
> - separate evaluation under full, lab-only, and field-only regimes, isolating domain effects,
> - explicit quantification of domain-induced performance gaps,
> - structured failure-case inspection.
>
> We agree that mechanistic analysis could be valuable, but it falls outside the scope of the current work.
>
> To provide deeper insight into model behaviour under domain shift, we analysed performance as a function of class-level domain composition using the best model from each architecture. Classes were grouped as balanced or lab-dominant based on training distribution and filtered by field support in the test set ($\geq$11 samples; max $=$23).
>
> On balanced classes, all models achieve similar lab performance (F1 $\approx$ 0.93–0.94), but differ in generalisation: CNN (field F1 0.820, gap 0.118), contrastive VLM (0.792, 0.147), and generative VLM (0.824, 0.106), indicating lower domain sensitivity for VLM-based models.
>
> From balanced to lab-dominant classes, degradation increases for all models, but substantially more for CNNs: domain gap increases by +0.146 (CNN) vs +0.085 (contrastive) and +0.076 (generative), with corresponding larger drops in field performance. This shows that CNNs are more sensitive to reduced field exposure, while VLMs exhibit more stable cross-domain behaviour, consistent with stronger reliance of CNNs on domain-specific visual features.
>
> This is explicitly stated in the paper’s contributions:
> (1) AgriPath-LF16 as a domain-aware benchmark,
> (2) a unified comparison across CNNs, contrastive VLMs, and generative VLMs, and
> (3) deployment-context analysis under domain-focused training regimes.
>
> The contribution is therefore empirical and evaluative, not methodological.
>
> ---
>
> ### 3. Failure Analysis Specificity
> >“failure modes are generic (hallucination, parsing errors)”
> >“Make the failure analysis more specific to this task”
>
> While hallucination and parsing errors are general phenomena, here they manifest in task-specific forms tied to the crop–disease ontology. Specifically, we observe:
>
> - Ontology violations: generation of crop–disease pairs outside the dataset label space.
> - Invalid attribute composition: fabricated modifiers leading to non-existent classes
> - Ambiguous outputs: underspecified predictions that cannot be mapped to a unique class
> - Silent failures: empty generations representing breakdown under domain shift
>
> These arise from the interaction between free-text generation and a constrained agricultural label space under domain shift, rather than being generic LLM artefacts. We will clarify this more explicitly.
>
> ---

---

> ### Author Response · Authors · 2026-04-07
> **Response to Reviewer YEJb (Part 2)**
>
> ### 4. Scope Beyond the Dataset and Task
> >”evaluation is restricted to a single task… unclear how… conclusions generalize beyond this setting.”
> >”Discuss how well the conclusions generalize beyond this dataset… classification setting”
>
> The focus on fine-grained classification is intentional, as it isolates visual recognition under domain shift, which is a prerequisite for downstream agricultural systems. Prior benchmarks such as AgroGPT and MIRAGE operate in advisory or conversational settings and implicitly assume reliable visual perception; this work instead evaluates that underlying capability directly.
>
> Beyond the dataset, AgriPath-LF16 aggregates multiple sources into a unified ontology, spanning 16 crops and 65 unique crop-disease pairs across lab and field conditions. While this introduces variability (e.g., image quality, and potential similarity between samples), it reduces reliance on any single dataset bias. Crucially, the main trends—CNN sensitivity to domain shift vs VLM robustness—are consistent across controlled regimes (full, lab-only, field-only training), indicating they are not artefacts of a specific split or dataset configuration.
>
> That said, domain shift is operationalised here as lab vs field acquisition, and other factors (e.g., geography, temporal variation) are not explicitly captured. Accordingly, the findings are most directly applicable to visual diagnosis pipelines, and are expected to extend to multimodal systems that rely on visual grounding, while broader generalisation remains future work.
>
> We therefore agree that broader generalisation should be discussed more explicitly, but should remain as future work:
> - These findings are most directly applicable to visual diagnosis pipelines
> - They are expected to extend to multimodal systems that rely on visual grounding
>
> ---
>
> ### 5. Dataset Limitations and Impact on Results
> >”Add more discussion on dataset limitations (especially lab vs field imbalance) and how they might affect the results”
>
>
> While the AgriPath-LF16-30k subset reduces overall class imbalance, some crop-disease pairs remain lab-dominant due to limited field data availability.
>
> This affects evaluation in two ways. Firstly, class-level field performance becomes less reliable for underrepresented classes, and secondly, aggregate metrics may partially reflect performance on lab-dominant distributions.
>
> To address this, we explicitly analyse domain composition at the class level. Classes are grouped by training domain balance (balanced vs lab-dominant) and filtered by minimum field support (as seen in point 2), yielding a subset of classes where field evaluation is statistically meaningful. To summarise the analysis stated in point 2, the observed trends of CNN degradation versus VLM robustness persist within the filtered evaluable subset, indicating that our core conclusions are not driven by sparsity or imbalance artifacts.
>
> Finally, we emphasise that this imbalance is not purely incidental but reflects real-world data constraints, where field data is harder to collect. Retaining this structure enables evaluation under realistic deployment conditions, but necessitates careful interpretation of class-level results.
>
> We will incorporate this clarification into the Limitations & Future Work section to make the impact on evaluation more explicit.
>
> ---
>
> We would be happy to accept any additional suggestions that can improve our manuscript.

---

### Review · Reviewer_3YJv · 2026-03-25

**Summary Of Contributions:**

This paper studies architectural trade-offs for crop disease classification by comparing CNNs, contrastive VLMs, and generative VLMs under lab/field domain shift. It also introduces the AgriPath-LF16 benchmark and argues that model choice should depend on deployment context rather than aggregate performance alone. The paper’s main value is as an empirical comparison/benchmark paper rather than a methodological contribution.

**Additional Comments:**

Overall, I find the paper promising and potentially useful, especially as a benchmark-style empirical study. My main concerns are comparison fairness, missing ACC, unclear explanation of PSR / generative reliability, and the absence of an accessible anonymous dataset link. The draft also needs some presentation cleanup.

**Audience:**

Yes

**Audience Explanation:**

The paper should be of interest to readers working on robustness, domain shift, multimodal evaluation, and applied ML in agriculture. The lab/field split and deployment-oriented framing are useful, even though the contribution is primarily empirical.

**Broader Impact Concerns:**

The broader-impact section is relevant, but it would be stronger if it connected the observed failure modes of generative models to concrete mitigation strategies, such as constrained outputs, abstention, or human oversight.

**Claims And Evidence:**

Yes

**Claims Explanation:**

Yes, but only partially.

The main qualitative claims are broadly supported, but I have several concerns about clarity and experimental support:

- The comparison across model families does not seem fully fair, since the adaptation/tuning budgets differ substantially across CNNs, contrastive VLMs, and generative VLMs.
- The paper does not report accuracy (ACC), which would be a useful complementary metric in this multi-class classification setting.
- The notion of “generative reliability” is not clearly explained. It is unclear whether this refers to parsability, semantic correctness, or practical trustworthiness.
- PSR is reported but not clearly defined or motivated early enough in the paper.
- The generative MCQ setting is not directly comparable to the other classification settings and should be interpreted more carefully.
- The paper mentions dataset release, but I could not find a working anonymous dataset link, which limits reproducibility during review.

**Requested Changes:**

- Clarify and justify the fairness of the comparison across model families.

- Add ACC alongside macro-F1/precision/recall.

- Clearly explain what is meant by **generative reliability**.

- Define PSR explicitly at first use and explain why it is needed.
- Provide a functioning anonymous link to the dataset/resources.

- Clarify dataset construction details, including label harmonization, deduplication, and split protocol.

- Tone down any claims that go beyond what the experiments directly establish.
- Figure 1’s two subfigures look visually unbalanced.
- Figure 2 caption appears to be missing ending punctuation.
- Some figures/tables are far from the corresponding discussion, and it is not always clear whether they are in the main text or appendix.
- Figure 4 appears to contain an unexplained watermark and should be cleaned up.
- The overall presentation would benefit from another round of polishing.

---

> ### Author Response · Authors · 2026-04-07
> **Response to Reviewer 3YJv (Part 1)**
>
> We appreciate the useful feedback provided by the reviewer. We address the reviewer’s comments below:
>
> ---
>
> ### 1. Fairness of Comparison across Model Families
> >”Clarify and justify the fairness of the comparison across model families.”
>
> The goal is not to enforce identical training procedures, but to evaluate each paradigm under representative and commonly used adaptation regimes proposed in the literature:
> - CNNs: partial fine-tuning -- early layers frozen, with adaptation restricted to high-level representations (final residual block + classifier)
> - Contrastive VLMs: frozen encoders with linear probing
> - Generative VLMs: parameter-efficient fine-tuning (LoRA)
>
> These choices reflect practical usage while controlling adaptation capacity.
>
> Additionally, we have included the ConvNeXt [[Liu et al., 2022]](https://arxiv.org/abs/2201.03545) CNN model, a more recent CNN than ResNet-50. Within this model family, we have explored ConvNeXt-Tiny which is of a similar size as ResNet-50 (28.6M vs 25.6M). We report our findings in the table below..
>
> | Paradigm | F1(main) | F1(lab) | F1(field) |
> |---|---|---|---|
> | ResNet-50 | 91.0 | 97.2 | 69.9 |
> | ResNet-50-Lab | 56.4 | 96.7 | 4.3 |
> | ResNet-50-Field | 37.1 | 13.4 | 73.2 |
> | ConvNeXt-Tiny | 92.9 | 97.1 | 74.6 |
> | ConvNeXt-Tiny-Lab | 56.9 | 97.5 | 6.3 |
> | ConvNeXt-Tiny-Field | 40.1 | 12.8 | 78.8 |
>
> Finally, we also conducted hyperparameter sweeps for each model (currently in the appendix), and selected the run with the lowest validation loss to report in the main body of the paper. This is to ensure strong configurations for each model family. We will include this clarification.
>
> ---
>
> ### 2. Accuracy (ACC) not reported
> >”The paper does not report accuracy (ACC)...”
>
> We agree that while accuracy is a useful complementary metric, given the nature of the dataset, the balanced accuracy metric overlaps with the macro-recall metric. Balanced accuracy is defined as the average of per-class recalls, which is the same definition of macro-recall. We computed balanced accuracy during evaluation and observed that it is equivalent to macro-recall across all experiments. Below is a table comparison of the two metrics for the set of experiments trained and evaluated on the full dataset.
>
> | Model | Balanced Accuracy | Macro-Recall |
> |---|---|---|
> | ResNet-50 | 90.71 | 90.71 |
> | ConvNeXt-Tiny | 92.99 | 92.99 |
> | SigLIP | 90.05 | 90.05 |
> | CLIP-L-14 | 91.04 | 91.04 |
> | SmolVLM-500M | 87.53 | 87.53 |
> | Qwen2.5-VL 3B | 88.78 | 88.78 |
> | Qwen2.5-VL 7B | 90.50 | 90.50 |
>
> We will make sure that the section that reports all our evaluation metrics reports this additional clarification to avoid any confusion.
>
> ---
>
> ### 3. Generative Reliability
> >”The notion of “generative reliability” is not clearly explained.”
>
> We agree that this term was underspecified.
>
> In the paper, “generative reliability” refers specifically to output parsability under task constraints (PSR) rather than correctness. We will clarify this distinction explicitly in our experimental evaluation section.
>
> ---
>
> ### 4. PSR definition and motivation
> >”PSR is reported but not clearly defined or motivated early enough in the paper.”
>
> We agree that Parse Success Rate (PSR) should be defined earlier.
>
> PSR measures the fraction of outputs that can be mapped to a valid label in the crop–disease taxonomy, capturing format compliance / parsability of generative outputs.
>
> This metric addresses a failure mode not captured by standard metrics such as F1, which assume valid predictions. Since PSR and F1 capture complementary dimensions of behaviour (PSR is near-perfect but F1 is low under domain shift), the motivation of this metric is that structural reliability and semantic correctness are not aligned
>
> We will revise the paper to include this explanation in the experimental evaluation section.
>
> ---
>
> ### 5. Dataset link
>
> We apologise for the issue with the anonymous dataset link. The dataset size (~30k images) limits direct hosting within the review system.  We will include a representative subset in the supplementary material. The subset will contain 260 samples in total, with each class having 4 samples (2 lab and 2 field where applicable).
>
> ---

---

> ### Author Response · Authors · 2026-04-07
> **Response to Reviewer 3YJv (Part 2)**
>
> ### 6. Dataset construction details
> We agree that additional clarity would be helpful.
>
> **Label harmonisation:**
> Images from multiple public datasets were mapped into a unified crop–disease ontology, with each class defined as a canonical crop–disease pair. This ensures consistent labels across heterogeneous sources.
>
> **Domain annotation:**
> Each image is explicitly assigned a domain label (lab or field) based on its source dataset. This annotation is used throughout to enable controlled domain-specific evaluation.
>
> **Deduplication:**
> We conducted a post-hoc duplicate analysis using image-level hashing and identified a small overlap (less than 6.5% instances were affected by this). Re-running experiments after resolving the overlap resulted in minimal differences in performance, and key trends reported in the paper remain supported. We add this additional filtering step to our dataset preparation protocol.
>
> **Split protocol:**
> The dataset is split into train/validation/test (80/10/10) with class-level stratification over crop–disease pairs, ensuring consistent class coverage across splits.
>
> We will include the above in addition to existing data composition details.
>
> ---
>
> ### 7. Claims beyond experimental evidence
> > “Tone down any claims that go beyond what the experiments directly establish.”
>
> We agree that some forward-looking statements may be overly strong. For example:
>
> - *“Additionally, integrating contextual metadata into generative VLM pipelines may improve reliability and reduce hallucinations through agricultural grounding”* **revised to** *“Additionally, integrating contextual metadata into generative VLM pipelines would provide a look into capitlising on the multimodal capabilities of VLMs”*
> - *“Moreover, reasoning-aware fine-tuning that promotes localisation of symptomatic regions and suppresses background cues may improve robustness”* **revised to** *“Moreover, reasoning-aware fine-tuning that promotes localisation of symptomatic regions and suppresses background cues would allow for further exploration into the use of different architectures when translating across domains with varying visual environmental noise”*
>
> We will revise them to clearly reflect future work or hypotheses rather than conclusions.
>
> ---
>
> ### 8. Formatting and presentation
> We thank the reviewer for highlighting these issues and will address all formatting inconsistencies in the revision.
>
> - Re Figure 1: We have adjusted the sizes of the subfigures to fixed heights.
> - Re Figure 2: We have added the missing punctuation here.
> - Re Figures/tables being far from discussion: We will improve placement and referencing for clarity. Specifically, we will ensure that key figures appear close to their first mention, and explicitly indicate when content is located in the appendix (e.g., “see Appendix Table X”), to remove ambiguity about placement.
> - Re Figure 4: We will swap the image to one without a watermark. The AgriPath dataset includes data from the PlantDoc dataset (as mentioned in the appendix table) which had the watermarked image. Upon manual inspection there are a few instances of these, although not enough to be problematic.
> - Re General Polish: We will perform an additional round of presentation refinement, including improving clarity of explanations, ensuring consistent terminology, tightening figure/table referencing, and correcting minor grammatical issues throughout.
>
> ---
>
> ### 9. MCQ setting not directly comparable
> >”The generative MCQ setting is not directly comparable…”
>
> We appreciate this point. The MCQ setting is not intended as directly comparable to free-text generation, but rather as a progressive constraint on the output space of zero-shot experiments to reduce parsing ambiguity.
>
> The zero-shot experiments for generative models follow a structured progression:
> - Pure ZS: unconstrained generation
> - Context-aware ZS: full label space provided
> - MCQ ZS: restricted candidate set (4 options)
>
> This reflects increasing levels of guidance, motivated by examining inherent model capability without output controllability and parsing reliability constraints. We will clarify this and ensure MCQ is interpreted as a controlled intervention rather than a directly equivalent comparison.
>
> ---
>
> We welcome any specific suggestions that would meaningfully strengthen the manuscript.

---

### Review · Reviewer_ckCa · 2026-03-25

**Summary Of Contributions:**

This paper presents a systematic empirical comparison of three model paradigms for crop disease classification: CNNs (ResNet-50), contrastive VLMs (CLIP, SigLIP), and generative VLMs (Qwen2.5-VL, SmolVLM). To support controlled evaluation, the authors introduce AgriPath-LF16, a benchmark of 111k images spanning 16 crops and 41 diseases with explicit separation between laboratory and field imagery, along with a balanced 30k subset. Models are evaluated under full, lab-only, and field-only training regimes using macro-F1 and Parse Success Rate (PSR).

**Additional Comments:**

### Strengths

1.  **Well-Designed Experimental Protocol:** The unified evaluation across three distinct model paradigms under matched training regimes (full, lab-only, field-only) is a sound experimental design that enables fair cross-architecture comparison. The inclusion of PSR as a metric for generative models is a thoughtful addition.
2.  **Practical Dataset Contribution:** AgriPath-LF16 addresses a genuine gap by providing explicit lab/field domain labels across 16 crops and 41 diseases. The balanced 30k subset with documented downsampling rules facilitates reproducible benchmarking.
3.  **Informative Failure Analysis:** The qualitative analysis of failure modes across model families (Section 4.5) provides useful insights, particularly the characterisation of generative VLM hallucinations (e.g., fabricated crop names, underspecified labels).

### Weaknesses

The paper makes a reasonable empirical contribution but has several weaknesses that limit its significance and novelty.

1.  **Limited Model Coverage** Only a single CNN architecture (ResNet-50) is evaluated. Modern CNN variants such as ConvNeXt [1] and EfficientNet [2] would provide a much fairer representation of the CNN paradigm. ResNet-50 is no longer representative of the state of the art for CNNs, and its poor cross-domain performance may reflect architectural limitations specific to ResNet rather than CNNs as a class.


2.  **Missing Domain Adaptation Baselines:** The paper frames domain shift as a central challenge but does not compare against any domain adaptation or domain generalisation techniques (e.g., DANN [3], MixStyle [4], or simple data augmentation strategies targeting domain gap). Without these baselines, it is unclear whether the observed robustness differences stem from architectural properties or could be addressed with straightforward techniques applied to any architecture.

3.  **Limited Novelty in Findings:** The core conclusions (CNNs overfit to training domain, VLMs with large-scale pretraining generalise better) are well-established in the transfer learning and domain adaptation literature. The paper would benefit from deeper analysis that yields new insights beyond confirming known trends in an agricultural setting.

4.  **Dataset Construction Concerns:**
    *   The dataset is assembled from existing public datasets (PlantVillage, PlantDoc, etc.) rather than being a new data collection effort. While aggregation has value, the paper does not sufficiently discuss potential issues such as label inconsistency across sources, image quality variation, or duplicate/near-duplicate images between source datasets.
    *   Some crop-disease pairs appear to have very few field samples even after balancing (visible in Figure 2), which could make the field evaluation unreliable for those classes.


[1] "A ConvNet for the 2020s." CVPR (2022).

[2] "EfficientNet: Rethinking Model Scaling for CNNs." ICML (2019).

[3] "Domain-Adversarial Training of Neural Networks." JMLR (2016).

[4] "Domain Generalization with MixStyle." ICLR (2021).

**Audience:**

Yes

**Audience Explanation:**

N/A

**Claims And Evidence:**

No

**Claims Explanation:**

Check weakness please.

**Requested Changes:**

1.  **Broaden CNN baselines** to include at least one modern architecture (e.g., ConvNeXt) to ensure the CNN paradigm is fairly represented.
2.  **Add domain adaptation baselines** to contextualise whether the observed domain shift can be mitigated with existing techniques.
3.  **Discuss dataset assembly limitations**, including potential label noise, cross-source inconsistencies, and the reliability of evaluation on classes with very few field samples.

---

> ### Author Response · Authors · 2026-04-07
> **Response to Reviewer ckCa (Part 1)**
>
> We thank the reviewer for the detailed feedback and for highlighting several important considerations. We address each point below:
>
> ---
>
> ### 1. Limited Model Coverage
> >”Modern CNN variants such as ConvNeXt and EfficientNet would provide a fairer representation”
>
> We agree that including more recent CNN architectures would strengthen the evaluation.
>
> ResNet-50 was selected as a widely used and well-established baseline in prior agricultural and transfer learning literature. We acknowledge that more modern architectures (e.g., ConvNeXt) may provide a stronger reference point.
>
> Preliminary ConvNeXt findings:
> Using ConvNeXt-Tiny, we ran training and evaluation exactly as we did for ResNet-50, and selected the parameter-set to discuss in the paper according to the lowest validation loss (as was done with ResNet-50).
>
> | Paradigm | F1(main) | F1(lab) | F1(field) |
> |---|---|---|---|
> | ResNet-50 | 91.0 | 97.2 | 69.9 |
> | ResNet-50-Lab | 56.4 | 96.7 | 4.3 |
> | ResNet-50-Field | 37.1 | 13.4 | 73.2 |
> | ConvNeXt-Tiny | 92.9 | 97.1 | 74.6 |
> | ConvNeXt-Tiny-Lab | 56.9 | 97.5 | 6.3 |
> | ConvNeXt-Tiny-Field | 40.1 | 12.8 | 78.8 |
>
> While ConvNeXt does show more promising generalisation than ResNet-50 when given field data to train on, we observe that when completely removing the domain from the train set, ConvNeXt collapses similarly to ResNet-50. When given only field data upon training, ConvNeXt shows the best in-domain performance across all models, but also has the second-to-worst performance in out-of-domain performance, even below ResNet-50. These findings confirm known trends that CNNs are superior than other architectures when being utilised on in-domain data according to their training data.
>
> We will share these findings to represent the current CNN paradigm better.
>
> ---
>
> ### 2. Missing Domain Adaptation Baselines
> >”no comparison against DANN, MixStyle, etc.”
>
> We would like to thank the reviewer for the suggestion.
>
> The goal of the paper is to compare model families under standard and representative adaptation regimes, rather than to evaluate domain adaptation techniques themselves. Methods such as DANN or MixStyle introduce additional, architecture-specific modifications, primarily applicable to CNNs, and are not directly transferable to contrastive or generative VLMs.
>
> Including such techniques would therefore break cross-paradigm comparability and shift the paper toward a methodological study of domain adaptation, which was not our objective.
>
> Instead, the paper evaluates how different architectures behave under domain shift without specialised adaptation, which we view as a practically relevant setting.
>
> ---

---

> ### Author Response · Authors · 2026-04-07
> **Response to Reviewer ckCa (Part 2)**
>
> ### 3. Limited Novelty in Findings
> >“The core conclusions… are well-established in the transfer learning and domain adaptation literature.”
>
> We agree that some behaviours (e.g., domain sensitivity of CNNs, robustness of large pretrained models) are known in isolation.
>
> The contribution of this work is not to claim novel behaviours, but to provide a controlled, cross-paradigm evaluation of these behaviours within a single, domain-explicit benchmark (AgriPath-LF16), under matched conditions. Existing work evaluates these model families either in isolation or on datasets that do not support controlled domain-shift analysis (e.g., single domain-only or crop-focused datasets). This paper enables a controlled, domain-explicit comparison across CNNs, contrastive VLMs, and generative VLMs across a variety of crops and diseases.
>
> This setting is non-trivial because fine-grained crop–disease classification under lab–field variation introduces systematic visual shifts (background clutter, lighting, lesion variability) that are not captured in standard benchmarks.
>
> **Re: Deeper Analysis**
> >“... benefit from deeper analysis that yields new insights…”
>
> We agree that deeper analysis strengthens the contribution, and we provide additional class-level analysis below.
>
> The current paper already includes:
> domain-specific evaluation (full vs lab vs field),
> quantification of robustness gaps,
> and structured failure analysis.
>
> To further strengthen this, we analysed class-level effects under dataset imbalance for the best model from each architecture. Classes were grouped by training domain composition (balanced vs lab-dominant) and filtered by field support in the test set ($\geq$ 11 samples; max $=$ 23).
>
> On balanced classes, all models achieve similar lab performance (F1 $\approx$ 0.93–0.94), but differ in generalisation: CNN (field F1 0.820, gap 0.118), contrastive VLM (0.792, 0.147), and generative VLM (0.824, 0.106). This shows that generative VLMs experience the least domain degradation, albeit minimal.
>
> From balanced to lab-dominant classes, degradation increases for all models, but substantially more for CNNs: domain gap increases by +0.146 (CNN) vs +0.085 (contrastive) and +0.076 (generative), with corresponding larger drops in field F1. This indicates that CNNs are more sensitive to reduced field exposure, while VLMs exhibit more stable cross-domain behaviour.
>
> This provides more granular insight while remaining within the scope of a diagnostic empirical study. We will add this analysis as a section in our experimental evaluation to complement our error analysis.
>
> ---
>
> ### 4. Dataset Construction Concerns
>
> We agree that these are important considerations and clarify them below.
>
> Label harmonisation: Source datasets were mapped into a unified crop–disease ontology with consistent class definitions.
>
> Image quality variation: Variation in quality and acquisition conditions is inherent to real-world agricultural data and is part of the domain shift that our benchmark is designed to capture.
>
> Duplicates / near-duplicates: We conducted a post-hoc duplicate analysis using image-level hashing and identified a small overlap (less than 6.5% instances were affected by this). Re-running experiments after resolving the overlap resulted in minimal differences in performance, and key trends reported in the paper remain supported. We add this additional filtering step to our dataset preparation protocol.
>
> We will incorporate these clarifications into the Dataset Methodology section.
>
> **Re: Field Evaluation Unreliability**
> >“field evaluation may be unreliable for some classes”
>
> As noted in the paper, some crop–disease pairs remain underrepresented in the field domain due to data availability.
>
> Regarding the actual data for this analysis, classes were grouped by training domain composition (balanced vs lab-dominant) and filtered by field support in the test set ($\geq$11 samples; max $=$23). This yields 12 balanced and 8 lab-dominant classes with reliable field evaluation; no extremely lab-dominant classes meet this criterion, indicating that class-level field performance cannot be reliably estimated in that regime. Field support in the lab-dominant group of the test set remains moderate (11–17 samples), indicating this trend is not driven by extreme sparsity.
>
> To better characterise this effect, we will extend the Evaluation & Analysis section with per-class analysis as discussed above.
>
> ---
>
> We remain open to any concrete recommendations that would further enhance the clarity and quality of the manuscript.

---

### Author Response · Authors · 2026-04-17
**Summary of Revisions in the Updated Manuscript**

Following the submission of the revised manuscript, we provide a concise summary of the key changes made in response to reviewer feedback for ease of reference.

---

- **Deeper Analysis:** Added a per-class, domain-constrained analysis to provide more granular insight into cross-domain behaviour across model families.
> Location of change: Section 4 - Evaluation & Analysis
- **Failure analysis clarification:** Refined the analysis of generative VLM failure modes to highlight task-specific manifestations (e.g., ontology violations, ambiguous outputs) rather than generic LLM behaviour.
> Location of change: Section 4.6.3 - Generative VLM Failure Modes
- **Scope and limitations:** Expanded discussion of generalisation beyond the dataset and task, and strengthened the limitations section to explicitly address lab–field imbalance and its impact on evaluation.
> Location of change: Section 5.1 - Limitations & Future Work
- **Dataset construction transparency:** Added details on label harmonisation, domain annotation, and dataset split protocol in the dataset subsection.
> Location of change: Section 3.1 - Dataset
- **Fairness of comparison:** Clarified adaptation strategies for each model family (CNN fine-tuning, contrastive linear probing, generative LoRA), included results for a modern CNN baseline (ConvNeXt-Tiny), and added details on hyperparameter sweeps.
> Location of change: Section 3 - Methodology; Section 4 - Evaluation & Analysis
- **Metric clarification:** Clarified the use of macro-averaged metrics, explicitly defined “generative reliability” as output parsability, and introduced Parse Success Rate (PSR) earlier with clear motivation.
> Location of change: Abstract; Section 4 - Evaluation & Analysis
- **Zero-shot protocol clarification:** Revised the description of generative VLM zero-shot experiments to clarify the progression from unconstrained generation to MCQ-based constrained evaluation.
> Location of change: Section 3.2.3 - Generative VLM Experiments
- **Claims and interpretation:** Revised forward-looking statements to better distinguish empirical findings from hypotheses and future work.
> Location of change: Section 5.1 - Limitations & Future Work
- **Presentation improvements:** Addressed formatting and clarity issues, including figure balancing, removal of watermarks, improved figure/table placement, and general language polishing.
> Location of change: General paper polishing
- **Dataset availability:** Included a representative subset of the dataset in the supplementary material to support reproducibility during review.
> Location of change: Supplementary Material

---

### Decision · Action_Editor_LSpb · 2026-04-20

**Recommendation:** Accept as is

**Audience:**

Yes

**Audience Explanation:**

This paper studies architectural trade-offs for crop disease classification by comparing CNNs, contrastive VLMs, and generative VLMs under lab/field domain shift. The topic could be interesting to the TMLR audience.

**Claims And Evidence:**

Yes

**Claims Explanation:**

This paper studies architectural trade-offs for crop disease classification by comparing CNNs, contrastive VLMs, and generative VLMs under lab/field domain shift. After the author-reviewer discussions, the reviewers believe the paper is a solid, well-executed empirical study that compares different model families under domain shift. The authors did a good job addressing most of the review comments, especially around baselines, metrics, and dataset details, and overall, the claims are reasonably well supported by the experiments. However, the novelty is somewhat limited. Based on the TMLR guidelines, I tend to accept this paper, but it doesn't qualify for the NeurIPS/ICLR/ICML Journal-to-Conference Track.